# Supramolecular Chirality in Azobenzene-Containing Polymer System: Traditional Postpolymerization Self-Assembly Versus In Situ Supramolecular Self-Assembly Strategy

**DOI:** 10.3390/ijms21176186

**Published:** 2020-08-27

**Authors:** Xiaoxiao Cheng, Tengfei Miao, Yilin Qian, Zhengbiao Zhang, Wei Zhang, Xiulin Zhu

**Affiliations:** State and Local Joint Engineering Laboratory for Novel Functional Polymeric Materials, Jiangsu Key Laboratory of Advanced Functional Polymer Design and Application, College of Chemistry, Chemical Engineering and Materials Science, Soochow University, Suzhou 215123, China; studentscheng@163.com (X.C.); 20194009023@stu.suda.edu.cn (T.M.); ylqian@stu.suda.edu.cn (Y.Q.); zhangzhengbiao@suda.edu.cn (Z.Z.); xlzhu@suda.edu.cn (X.Z.)

**Keywords:** supramolecular chirality, Azo-containing polymers, postpolymerization self-assembly, in situ polymerization-induced self-assembly

## Abstract

Recently, the design of novel supramolecular chiral materials has received a great deal of attention due to rapid developments in the fields of supramolecular chemistry and molecular self-assembly. Supramolecular chirality has been widely introduced to polymers containing photoresponsive azobenzene groups. On the one hand, supramolecular chiral structures of azobenzene-containing polymers (Azo-polymers) can be produced by nonsymmetric arrangement of Azo units through noncovalent interactions. On the other hand, the reversibility of the photoisomerization also allows for the control of the supramolecular organization of the Azo moieties within polymer structures. The construction of supramolecular chirality in Azo-polymeric self-assembled system is highly important for further developments in this field from both academic and practical points of view. The postpolymerization self-assembly strategy is one of the traditional strategies for mainly constructing supramolecular chirality in Azo-polymers. The in situ supramolecular self-assembly mediated by polymerization-induced self-assembly (PISA) is a facile one-pot approach for the construction of well-defined supramolecular chirality during polymerization process. In this review, we focus on a discussion of supramolecular chirality of Azo-polymer systems constructed by traditional postpolymerization self-assembly and PISA-mediated in situ supramolecular self-assembly. Furthermore, we will also summarize the basic concepts, seminal studies, recent trends, and perspectives in the constructions and applications of supramolecular chirality based on Azo-polymers with the hope to advance the development of supramolecular chirality in chemistry.

## 1. Introduction

Chirality is ubiquitous and can be observed at various hierarchical levels from subatomic and molecular to supramolecular and macroscopic scales [1,2,3]. As chiral phenomena widely exist in nature (such as the *D*-sugars as the main components of DNA and the secondary *R*-helix structure of proteins) and various artificial and biomimetic systems [4,5], they are very important not only for fundamental sciences but also for practical applications in many areas, such as molecular and chiral recognition [6], chiral sensors [2], chiroptical switches [7] and asymmetric catalysis [8]. The studies of chirality in terms of mechanisms and superstructures have always attracted significant attention from scientists. Indeed, from the understanding of chirality phenomena in natural systems to the increasing application to artificial intelligence systems, subjects related to chirality will clearly become more developed and provide broader guidance for academic research and industrial application [9,10,11].

Chirality has been regarded as a basic characteristic of living organisms and nature, which can be manifested at various hierarchical levels (Figure 1) [12]. The research on the mechanism of induction, transfer and amplification of chirality plays an important role in the evolution of nature, the origin of life and the development of chiral materials [13]. At the subatomic level, left-handed helical neutrinos indicates that chirality is related to parity conservation. Covalently bonded molecules can be chiral by asymmetrically arranging atoms in space around a center (point chirality), axis (axial chirality) or plane (planar chirality) [2]. The deeper investigations of molecular chirality can provide guidance for the design of drugs and functional materials. Furthermore, chirality also can be expressed at the supramolecular level (such as proteins and DNA) [14,15,16]. Supramolecular chirality involves the nonsymmetric arrangement of building blocks in a noncovalent self-assembly [17]. It should be noted that although supramolecular chirality is strongly related to the chirality of its components, however, it is not necessary that all building blocks have chiral nature. In most cases, supramolecular chirality can be constructed from chiral or achiral building blocks by weak noncovalent interactions, such as π-π stacking, hydrogen bonding, electrostatic interactions, van der Waals force and so on [18,19,20,21,22]. Chirality at the macroscopic level, such as the helical structure of cucumber vines and the spiral movement of the cosmic nebula, provides a powerful inspiration for mankind to manufacture controllable macroscopic helical materials. Among these various levels described above, supramolecular chirality is of vital importance because it is closely related to chemistry, physics, biomedical sciences and nanomaterials [23]. Chirality at a supramolecular level has attracted great attention due to rapid developments in supramolecular chemistry and self-assembly.

It is well known that self-assembly plays an important role in chemistry due to the rapid developments of supramolecular chemistry and nanotechnology. Generally, the term “self-assembly” can be defined as the process of autonomous arrangement of the system’s components into predefined architectures under given boundary conditions [24], which is a “bottom-up” approach to produce intriguing nanostructures [25]. In general, two approaches for preparing nanoassemblies are usually considered in the field of self-assembly chemistry [26]. The first one is the most commonly used method called postpolymerization self-assembly, which usually requires multiple-step procedures (e.g., polymerization followed by self-assembly by solvent or pH switch, dialysis or film rehydration) [27]. The first study highlighting the postpolymerization self-assembly in solution was reported by Eisenberg’s group [28], in which nanoparticles with different morphologies were produced by self-assembly of amphiphilic polystyrene-*b*-polyacrylic acid diblock copolymers using co-solvency manipulation. Using postpolymerization self-assembly, supramolecular nanoassemblies, both in solution and in film states, can be obtained by changing the external environmental factors [29]. Supramolecular self-assembly of polymers guided by traditional postpolymerization self-assembly offers a common strategy to produce high-performance, stimuli-responsive nanomaterials using weak noncovalent interactions [30]. The second method, named in situ supramolecular self-assembly, can be mediated by polymerization-induced self-assembly (PISA) [31] and enables the precise preparation of amphiphilic copolymer assemblies in situ during the process of polymerization [32]. PISA-mediated in situ supramolecular self-assembly strategy is a powerful and convenient tool for constructing desired supramolecular morphologies without any post-polymerization processing steps.

Self-assembly has been widely employed in constructing supramolecular assemblies of Azo-polymers. Azo-polymers are very promising in the field of reversible photo-switchable materials [33,34,35]. With external stimuli, the Azo units in supramolecular Azo-polymers can dramatically change their polarity, shape, and sizes, which can be easily detected by UV-vis and circular dichroism (CD) spectroscopies. As a result, these polymers have found great applications in stimuli-responsive chiroptical switches [36]. In this review, we intend to discuss the supramolecular chirality of Azo-polymer systems, constructed by traditional postpolymerization self-assembly and PISA-mediated in situ supramolecular self-assembly strategies, for the consideration of successfully understanding and applying supramolecular chirality in stimulus-responsive materials. First, we will present an introduction to the basic concepts of supramolecular chirality based on Azo-polymers, along with some commonly used characterization techniques to examine and understand these supramolecular systems. Then, we will summarize the seminal studies and recent developments in the constructions and applications of supramolecular chirality based on Azo-polymers. Finally, we will focus on the perspective of current limitations along with future opportunities for the further advancement of supramolecular chirality in Azo-polymers.

## 2. Basic Concepts Related to Supramolecular Chirality

Chirality usually describes objects that exist as a pair of non-superimposable mirror images. Molecular asymmetry is the root of chirality. As we know, chirality at the *molecular level* is exhibited by the unique arrangements of atoms in space. Molecular chirality can be basically classified as point, axis, and plane chirality, according to the differences of each chiral feature [4]. The components in molecules asymmetrically arranged in space around a center, axis, or plane are called point, axial, and planar chirality, respectively. Supramolecular chemistry has received significant attention from scientists and experienced great development since it was proposed by Lehn et al. in 1978 [37]. Chirality expressed at the *supramolecular level* can be classified as supramolecular chirality [14]. Supramolecular chirality involves the nonsymmetric arrangement of building blocks in a noncovalent assembly. Generally, chirality can be determined by the properties of the components. Thus, chirality can be divided into different levels based on its constituents. First-level chirality refers to the asymmetry of atoms. Second-level chirality refers to the chirality generated by the asymmetric configuration or conformation of chiral molecules. Third-level chirality refers to the chirality of supramolecular aggregates formed by weak noncovalent bonds between molecules. Fourth-level chirality refers to more complex chiral structures obtained by the combined action of the first three levels of chiral structures. Here, some basic concepts of characterization techniques for molecular and supramolecular chirality systems based on Azo-polymers will first be discussed.

### 2.1. Configurational, Conformational and Helical Chirality

Chiral structures can be constructed by not only covalently bonded molecules with defined configuration and conformation, but also noncovalent bonded supramolecular structures with conformational flexibilities. Configuration refers to the permanent geometry of molecules with spatial arrangement connected by covalent bonds, while conformation refers to the different arrangement of atoms (or groups) in molecules due to free bond rotation under certain conditions. Configurational chirality (Figure 2a) comes directly from the covalent arrangement of the molecular structure, which cannot be changed without breaking the covalent bond [38]. The most basic form of configuration chirality is point chirality. Point chirality involves four different substituents bonded to the central atom, resulting in a molecule entirely non-superimposable on its mirror image. Conformational chirality derives from asymmetric rearrangements in molecules or assemblies that removes any elements of symmetry (Figure 2b) [39]. In contrast, the isomerization of conformational enantiomers can occur either by bond rotation or bond angle inversion under the external chiral stimulus [40]. Generally, conformational chirality in polymer systems usually involves helical chirality or supramolecular chirality, which may be maintained by steric hindrance or weaker noncovalent interactions.

Helical chirality (Figure 2c) is a special kind of conformational chirality, which is common and widely reported in supramolecular and polymer systems. Helical chirality can be constructed in molecules or assemblies with a unidirectional nature of the twist propagation along the axis [41]. The construction of enantio-enriched helices in supramolecular polymers has been also accomplished by the development of the “sergeants-and-soldiers” principle [42] or “majority rule” [43]. Many examples of helical chirality can be found in natural (DNA or *R*-helix of proteins) and artificial systems (helicenes [44], dipyrrins [45], polyacetylenes [46], poly(meth)acrylates [47] and polyisocyanides [48]). For example, helical chirality in polymer systems can be constructed by asymmetric polymerization using a chiral monomer, initiator, ligand or catalyst [49,50,51]. In 1979, Okamoto et al. reported the first synthesis of one-handed helical polymethacrylate with a nearly 100% isotacticity by helix-sense selective polymerization (HSSP) of an achiral bulky monomer using a chiral ligand [52]. In 2009, Wan et al. reported that the long-range chirality transfer can be achieved in the free radical polymerization of bulky vinyl monomers containing chiral side groups [53]. Wu et al. developed a novel synthetic strategy for the preparation of optically active helical polyisocyanides by using an achiral initiator in the presence of a chiral additive [54]. Recently, our group presented a helix-sense selective dispersion polymerization (HSSDP) strategy for preparing optically active monodisperse Azo-containing polymer particles with supramolecular chirality for the first time [55]. Generally, the term “*M*/*P* helicity” of helical chirality is usually used in helical systems [17]. The conventional identification method is that when looking from either end downward along the helical axis, the chirality is termed *P* helicity if the building blocks rotate clockwise, and termed *M* helicity if they rotate anticlockwise.

### 2.2. Supramolecular Chirality

Similarly, chirality expressed at the *supramolecular level* can be termed supramolecular chirality (Figure 3a). Supramolecular chirality can be constructed from the nonsymmetric arrangement of building blocks by weak noncovalent interactions. The building blocks may be chiral molecular components, achiral molecules or a combination of chiral and achiral molecules. Mostly, chiral molecular components are usually employed to generate supramolecular chiral structures through noncovalent weak interactions, such as π-π stacking, hydrogen bonding, electrostatic interaction and host-guest interaction [4], using either traditional postpolymerization self-assembly or in situ supramolecular self-assembly. However, it is not required that all components are chiral in supramolecular chiral structures, although the systems usually tend to follow the chirality of chiral sources in most case. Supramolecular chirality produced from achiral building blocks can be achieved and has attracted more and more attention. Several preferred methods such as circularly polarized light (CPL), chiral solvation, and chiral gelation, have been successfully employed to produce supramolecular chirality from achiral building blocks [56,57,58,59]. In general, supramolecular chirality largely depends on the weak noncovalent interactions and the assembly manner of the chiral/achiral building blocks, i.e., (1) one or more of the components are chiral that the assembly has determined supramolecular chiral structures; (2) or the achiral components associate in noncovalent interactions that the assembly adopt predominant one handed helicity. Generally, supramolecular chiral structures are dynamic and can be easily controlled by changing the external environment [60]. For example, this dynamic process can be regulated by varying the temperature and the concentration of solutions, or the polarity of medium [61,62,63]. The left-helix and right-helix of supramolecular chiral structures also can be readily converted to each other under external stimuli by destroying and reorganizing the chiral superstructure.

Furthermore, supramolecular chiral structures in polymer systems have been a fascinating interdisciplinary area of research due to the complexity of polymer systems. In polymer systems, supramolecular chiral structures can be constructed by the assembly of side-/main-chain chiral building blocks or achiral counterparts induced by chiral sources [4], which is usually called “induced chirality” in the latter case. Induced chirality of achiral assemblies is particularly common in supramolecular systems. It requires achiral guests to interact with chiral hosts through noncovalent interactions. In 1995, Yashima et al. reported the first helical polymer induced by chiral amines through acid-base interactions and systematically investigated the formation of one-handed helices from achiral polymers [64,65,66,67]. Fujiki et al. reported the transfer and amplification of chiral molecular information to achiral polysilylene aggregates through weak hydrogen bonding interaction [68]. Supramolecular chirality can be induced into achiral guests by asymmetric information transfer from chiral hosts. This asymmetry transfer process normally results in helical chirality that is maintained by weak noncovalent forces in supramolecular systems, even though the free building blocks are themselves still achiral (Figure 3). When achiral guests interact with chiral hosts, chiral information from chiral sources can be transferred to the achiral systems to form supramolecular chiral assemblies. Many chiral sources have been extensively studied including CPL [69,70], chiral additives [71,72,73], chiral solvents [74,75,76,77,78], chiral gelators [79,80,81], DNA [82], or chiral nanostructures [83,84,85,86]. 

It has been reported that several methods can be used in the characterization of chiral features of supramolecular systems. The absolute configuration of molecules can be determined by X-ray diffraction (XRD) [87], NMR spectroscopy [88] or stereo-controlled characterizations [89]. XRD is considered a reliable and convenient technique, which is useful for characterization of the absolute configuration of the chiral molecules. However, it requires a single crystal and typically at least one heavy atom [90]. XRD can give the basic chemical information of the component, such as the molecular geometry, bond distances and angles, and the packing of the molecules in the crystal [91]. So the molecular configuration and packing information obtained from the XRD structural analysis can help in understanding the self-assembly process. NMR is also an appealing technique for the determination molecular absolute configuration. Generally, two approaches are usually used to determine absolute configurations by NMR: (1) substrate analysis without derivatization and (2) analysis of the derivatives prepared from the substrate and the two enantiomers from a chiral derivatizing agent [92]. The second approach is widely selected because species produced by the substrate and the auxiliary reagent have a greater conformational rigidity, which in turn produces greater differences in the NMR spectra [88]. The stereo-controlled organic synthesis to prepare chiral substances usually has the disadvantages of being laborious and expensive.

The most commonly used methods for characterizing chirality are spectroscopic methods, such as circular dichroism (CD) and vibrational circular dichroism (VCD) spectroscopies. CD is the differential absorption of left versus right circularly polarized light [93]. The CD spectrum records the circular dichroism of the sample as a function of wavelength. CD technique is a powerful approach for detecting the characteristics of supramolecular chirality. The physical basis for the technique is the differential absorption of left and right circularly polarized light by substance in the absorbing region at UV or visible wavelengths [94]. The substance generally have chromophores for the CD signal to be detected in a certain wavelength range. Therefore, CD and UV-vis spectroscopies are often used and considered together in most supramolecular systems. VCD also offers a powerful approach for the detection of chiral properties which can be further combined with accurate quantum mechanical calculations [95]. Similar to CD, VCD spectrum records the differential absorption of a substance between the left and right circularly polarized light in the infrared region of the electromagnetic spectrum. VCD spectra can provide information about the absolute configuration of a chiral molecule. Furthermore, VCD spectroscopy has recently becoming more and more commonly combined with Fourier transform infrared spectrometry [95]. The peculiar and fascinating morphology of the supramolecular chiral structures can be directly observed with various microscopes. It is possible to directly characterize supramolecular chiral structures using scanning tunneling microscopy (STM), atomic force microscopy (AFM), scanning electron microscopy (SEM) and transmission electron microscopy (TEM) [4,96]. These morphological characterization technologies have greatly promoted the development of supramolecular chirality researches. Appropriate morphology characterization methods can be selected according to the characteristics of the samples.

### 2.3. Traditional Postpolymerization Self-Assembly versus In situ Supramolecular Self-Assembly

Self-assembly process involves “spontaneously” creating significantly ordered structures or patterns from disordered or homogeneous components. Nanostructures self-assembled by molecular or polymeric building blocks is ubiquitous in nature, with examples including DNA double helix and complex hierarchical structures of protein [97]. In the past few years, assemblies with multiple morphologies have found wide application in catalysis [98], nanoreactors [99], biomineralization [100], and drug delivery [101]. Traditional processes for generating polymer assemblies first requires the synthesis of polymers followed by assembly either by solvent or pH switch, dialysis, or film rehydration (Figure 4a) [102]. This method is called postpolymerization self-assembly, which has been widely studied by Eisenberg’s group [27,103,104,105]. Nevertheless, such protocols involve multiple steps and are typically conducted at highly diluted concentrations (≤ 1 wt% is common), which is difficult to implement on a large scale for industrial purposes. Furthermore, the process of supramolecular self-assembly in polymer systems is very fascinating but challenging due to the complexity of the components and interactions. Supramolecular self-assembly in polymer system can be defined as the spontaneous formation of organized structures from building blocks with the aid of weak noncovalent forces such as hydrophobic or solvophobic effect, π-π stacking, hydrogen bonding, electrostatic interactions and coordination interactions (Figure 4b) [29,106,107]. Previously, the processes and steps for obtaining supramolecular self-assemblies in polymer systems are similar to traditional postpolymerization self-assembly, in other words, the polymer needs to be synthesized before the self-assembly process. However, this strategy to produce supramolecular chiral assemblies in solution usually requires a relatively complex system consisting of the good solvent, poor solvent and chiral solvent, which can be very tedious. Hence, it is necessary to explore a more efficient and powerful synthetic methodology for precise and large-scale preparation of supramolecular assemblies.

Recently, a new strategy, called polymerization-induced self-assembly (PISA), emerged as an efficient technique for the preparation of polymer assemblies [108]. It has been widely recognized as a powerful platform for constructing amphiphilic block copolymer (BCP) nanostructures at high solid contents (up to 50 wt%) with the reliable control of spheres, worms, fibers, lamellae, vesicles and other unusual morphologies different from those obtained by the traditional postpolymerization self-assembly method, while without further postpolymerization processing steps. Generally, PISA is based on simultaneous polymerization mediated by controlled radial polymerization (CRP) and self-assembly process. PISA enables the precise and controllable preparation of polymer assemblies using “living”/controlled radical polymerization techniques, such as atom transfer radical polymerization (ATRP), nitroxide mediated polymerization (NMP), reversible addition-fragmentation chain transfer (RAFT) polymerization and click reaction [109]. Typically, RAFT-mediated PISA involves the chain extension of homopolymer (macromolecular chain transfer agent, macro-CTA) with co-monomers which form an insoluble second block as the polymerization proceeds. Self-assembly can be induced by the formation of the insoluble second block as the chain extends. The asymmetric amphiphilic BCPs obtained undergo spontaneous in situ self-assembly to form various nanostructures in the polymerization system. Meanwhile, the reaction media of PISA can be water, alcohol and nonpolar solvents, which have been extensively investigated by Armes [110], Boyer [111], Charleux [112], Pan [113], An [114], and others [115,116,117]. For examples, Zhou et al. reported that PISA strategy, with the advantages of in situ self-assembly, scalable preparation, and facile functionalization, can be used to prepare hierarchical multiscale sea urchin-like assemblies, taking advantage of epoxy thiol click polymerization [118]. The PISA process enables the effective preparation of nanostructures with varies morphologies starting with a single initial solvophilic homopolymer and extend the chain via one-pot reactions.

In addition to solvophilic/solvophobic interactions, many other factors can also greatly influence and promote PISA process during polymerization, such as liquid crystalline ordering [119], polyion complexation [120] and host-guest interactions [121], demonstrating the versatility of the PISA method. The influence factors can be roughly divided into the intrinsic and extrinsic factors, due to the difference in intrinsic architecture and nature of polymers and extrinsic conditions. The intrinsic factors include the degree of polymerization (DP), architecture and chemical nature of the copolymers, while the extrinsic factors cover conditions such as the solvent composition, the copolymer concentration and temperature. By varying these parameters, well-defined BCP nanostructures with diverse morphologies can be prepared by the PISA strategy [122]. Therefore, it is possible to employ the simple and efficient PISA strategy to prepare supramolecular assemblies in situ. Combining the advantages of PISA strategy with the characteristics of supramolecular chemistry, PISA-mediated in situ supramolecular self-assembly strategy will be a novel area of research and worth exploring (Figure 4c).

## 3. Supramolecular Chirality of Azo-Polymers Constructed by Postpolymerization Self-Assembly

Azo-polymers are promising reversible photo-switchable materials due to the unique change of mechanical and physicochemical properties [123,124,125,126,127,128]. Generally, a rod-like *trans*-form Azo unit can be isomerized to bent-shaped *cis*-form by UV light irradiation and reversibly transformed back to *trans* state by visible light or heat treatment, accompanied by changes in molecular configuration, polarity and other properties [129,130]. Furthermore, chirality in Azo-polymers has been widely investigated and supramolecular chirality has been frequently introduced to Azo-containing superstructures by asymmetric spatial arrangement of Azo building blocks [131,132,133]. Supramolecular chirality of side-chain Azo-polymers can usually be constructed by postpolymerization assembly using chiral Azo-polymers or using achiral Azo-polymers in which the chirality is induced by chiral sources [134,135,136]. Generally, the formation of the supramolecular chirality of Azo-polymers requires first the synthesis of the corresponding polymer, followed by self-assembly under selective conditions. Furthermore, the chirality of Azo-polymers can be disrupted by UV light irradiation, and dynamically recovered by visible light irradiation or temperature regulation owing to the reversible *trans*-*cis*-*trans* isomerization. Thus, the chiral-achiral chiroptical switches can be achieved based on the reversible *trans*-*cis* isomerization of Azo building blocks [137]. In this section, we will mainly discuss the construction of chirality in the side-chain photochromic Azo-polymers by postpolymerization self-assembly. It should be noted that all processes for building supramolecular chiral structures containing Azo units described in this section require at least two steps: synthesizing the corresponding polymers first and then performing the assembly.

### 3.1. Supramolecular Chirality of Chiral Azo-Polymers in Solution

Supramolecular chiral structures can be constructed directly by Azo-polymers with chiral groups in solution. Interestingly, the supramolecular chirality produced is significantly affected by the type of solvent and topological structure of Azo-polymers. Specifically, several examples will be discussed that represent the control over supramolecular chirality for self-assembled chiral Azo-polymer systems in solution. The detailed mechanism will be also discussed.

In 1974, conformational changes of side-chain Azo-polymers in solution were first observed by viscosity measurements [138]. Afterwards, Altomare et al. for the first time reported that the copolymer 1, consisting of Azo monomers with the optically active comonomer (-)-menthyl acrylate, exhibits chiroptical properties in the absorbance region of Azo chromophores. Initially, the chirality of the whole structure in solution was explained by considering the Azo side chains as disposed along chain sections with a predominant screw sense [139]. The presence of a CD couplet at 330–370 nm corresponding to the π-π* electronic transition is attributed to the relative dissymmetric disposition of the Azo chromophores. In 1991, copolymer 2 was further synthesized and investigated. The Azo side chains showed the induced conformational chirality. The irradiation-dependent chiroptical properties of the Azo side chains were limited by the spacer between the azo chromophores and main chain [140]. 

Angiolini et al. has reported many detailed studies on the mechanisms and chiroptical properties of Azo-polymer derivatives 3 based on chiral Azo monomers in dissolved state since 1998 [141,142,143,144,145]. The detection of chiroptical properties in chloroform solution suggested that the macromolecules existed in a conformational arrangement with a significant prevalence of one single chirality. This is caused by the effect of the stereogenic center on the growth of the polymer chain. In addition, significant changes in optical activity were also observed when other types of solvent (such as methanol) were added to the chloroform solution, thus suggesting the disturbance of a helical chiral macromolecular conformation in the polymeric backbone. Further investigations of chiral Azo-polymers in dissolved state have been reported by the same group [143,146,147,148,149,150]. In 1999, they described the syntheses and characterizations of new optically active Azo-polymer derivatives 4 containing a (*S*)-3-hydroxypyrrolidinyl group in solution [143]. CD and UV-vis spectra suggested that the presence of chirality in the Azo-polymers derived from conformational dissymmetry of the macromolecules and dipole-dipole interactions between the side-chain Azo chromophores. It should be noted that these works are preliminary explorations of chirality on the side-chain Azo-polymers in solution, which involves the ordered arrangement of azoaromatic chromophores in a helical geometry with one prevailing chirality. The Azo-polymer derivatives 5 also showed optical activity as demonstrated by the presence of strong exciton couplets in the CD spectra, with a mechanism similar to Azo-polymer derivatives 4 [151,152,153]. These fascinating works indicate that, in addition to the asymmetric chiral center present in the side group, the helical chirality of the Azo-polymer backbone can also be constructed in solution when the chiral center is close to the methacrylate main chain. Moreover, the helical structures of polymer backbones with predominantly one-handed screw sense in solution can be built directly by connecting the chiral Azo side chains to the backbone, as shown in 1,4-polyketones [154] and helical polyisocyanates systems [155,156].

Molecular-level chirality of Azo groups can induce an asymmetric arrangement of Azo units in Azo-polymers and result in the coexistence of molecular and supramolecular-level chirality in Azo-polymer aggregations. Oriol et al. synthesized and extensively studied Azo-polymers with chiral terminal tails, P4S and P8S (Figure 5), both in an aggregated state and in solution [136,157]. Although P8S contains chiral stereogenic center in the terminal alkyl chain, the polymer in dissolved state was completely CD-silent. However, significant bisignate CD signals of P8S aggregates could be detected in the dichloromethane (DCM)/hexane mixed system. In contrast to the conformational chirality reported by Angiolini [142], the chirality of P8S in aggregated state is associated with a supramolecular chiral assembly of Azo chromophores because the chiral terminal alkyl chain has a low influence on the polymer backbone [136,157]. In addition, the supramolecular chirality of the P8S polymer aggregates in DCM/hexane solution can be disrupted by the *trans*-*cis* photoisomerization of Azo units, which further confirmed the CD signals described above were from the supramolecular chirality of Azo stacks.

### 3.2. Supramolecular Chirality of Chiral Azo-Polymers in Film

The conformational or supramolecular chirality of chiral Azo-polymers in the solution state has been investigated in detail, as described above. However, the chiral induction, modulation, and memory of supramolecular chirality in polymer solid films are important in view of practical applications, such as chiral nonlinear optics, chiroptical switches and reversible optical storage. About 35 years ago, Todorov et al. reported an unexpected phenomenon that dichroism and birefringence can be observed from Azo-polymers in film when Azo chromophores undergo photoisomerization under polarized laser illumination [158]. It demonstrated the possibility of photomodulating the chiroptical properties of polymers containing optically active Azo chromophores. Afterwards, much effort was paid to chiral Azo-polymers (Figure 6) due to their particular optical properties, easy modulation of polymeric films and the stability of the chiral properties below *T*_g_. For instance, Kozlovsky et al. reported many outstanding works about chiral copolymers containing Azo units and mesogenic units with a chiral terminal chain. Photoinduced dichroism, optical anisotropy and photorecording properties in these chiral Azo-polymers 6, 7 had been triggered by the anisotropic angular distribution of *trans* and *cis* isomers [159,160,161,162]. Furthermore, a chiral Azo-homopolymer 8 with an Azo chromophore bearing a chiral terminal chain was also reported by their group, which exhibited a photoinduced birefringence property and smectic A (SmA) or a twist grain boundary A* (TGBA*)-like bistable phase depending on the thermal history of the polymer (in particular, the cooling rate from the isotropic state) [163,164,165]. It was reported that this bistable phase behavior is related to two different isotropic states. Similar chiral Azo-polymers with different spacer lengths and chiral responses in film that are manipulated by linearly polarized light (LPL) have been systematically investigated by Zhang and co-workers [166,167].

Bobrovsky et al. carried out comprehensive and detailed researches in terms of the characteristics and mechanism of the chiral Azo-polymers in film. For examples, in 2000, they reported a new type of multifunctional photochromic film based on dual photochromism of chiral Azo-copolymers 9 for optical data recording and storage. On the other hand, induced birefringence was observed when these polymer films were irradiated by polarized light, which was attributed to the isomerization of Azo groups and Weigert effect [168]. Bobrovsky and co-workers systematically investigated the thermo-, chiro- and photo-optical properties of cholesteric and chiral nematic Azo-copolymers in film, providing the possibility for the reversible as well as irreversible recording of optical information [169,170,171,172,173]. Furthermore, the photovariable supramolecular helical structure of chiral Azo-copolymer 10 can be constructed in film, providing high fatigue resistance and, therefore, allow a possible repeated (cyclic) “recording and erasing” of optical information [174,175]. In 2009, their group firstly synthesized and investigated a novel photochromic side-chain chiral Azo-homopolymer 11 with Azo fragment and chiral terminal chain. These types of polymer can form a smectic phase and a cholesteric supramolecular helical structure with selective light reflection in the IR region. In addition, partial homeotropic orientation, liquid crystalline ordering and further aggregation of the Azo groups in film by annealing procedures resulted in significant changes in UV-vis and CD spectra [176]. The photochromic side group structure, flexible spacer length, thermal treatment, and light irradiation significantly influenced the ordered arrangement of liquid crystal phase and supramolecular chirality (Figure 7) in thin spin-coated films of the chiral Azo-polymers [177,178,179,180]. These findings illustrate the mechanism of supramolecular self-assembly and drive the development of chiral materials based on Azo chromophores.

From Angiolini’s previous reports, it can be concluded that the chiral properties of Azo-polymers in solution arise from a helical chiral macromolecular conformation of the polymer backbone. In 2002, Angiolini and coworkers also reported the chiroptical properties of Azo-polymer derivatives 4 in film state [181]. The CD spectra of the chiral Azo-polymer films are similar to those of the polymers in solution, indicating that the Azo chromophores are also organized according to a chiral order in the solid state. However, the bisignate CD signal, both in solution and in solid thin films, is a typical behavior of exciton splitting determined by cooperative interactions between side-chain Azo chromophores arranged in one prevailing handedness. These results suggest that the macromolecules not only maintain chiral conformations with one prevailing helical handedness, but also have supramolecular interactions between side-chain Azo chromophores [182]. Besides, Natansohn et al. also presented the cooperative motion of two Azo structural units in film based on chiral Azo-copolymers 12 and 13 [132]. The results demonstrated the presence of supramolecular interactions between the neighboring side-chain Azo chromophores along the polymer backbone, which is similar to the previous results from Angiolini and coworkers. Taken together, many excellent works have been published for chiral Azo-polymers in the solid film [183,184,185,186,187,188], which further enhance our understanding of supramolecular chirality and offer many opportunities to develop new functional materials.

### 3.3. Supramolecular Chirality of Achiral Azo-Polymers

Even though supramolecular chirality can be directly constructed by chiral Azo-polymers both in solution and in solid film, the expensive synthesis and the limited types of chiral components are inevitable drawbacks. Therefore, the induction of supramolecular chirality from achiral Azo-polymer has great significance. Many strategies, such as chiral additives, chiral solvent and circularly polarized light (CPL) have been reported to induce supramolecular chirality from achiral Azo-polymers.

#### 3.3.1. Supramolecular Chirality Induced by Chiral Additives

Among these strategies, chiral additives, as molecules or polymers, have been used as chiral sources to induce supramolecular chirality and/or helical architectures from achiral Azo-polymers after postpolymerization processes. For example, Bobrovsky et al. extensively studied the use of chiral molecules (HexSorb and Sorb in Figure 8) as a source of chiral information to induce supramolecular chirality in achiral Azo-polymers 14, 15 and 16 [171,189,190,191,192,193]. The results showed that the films possess a noticeable circular dichroism with a maximum corresponding to the absorbance of the achiral Azo chromophores, suggesting that the supramolecular chirality was induced by chiral molecules in achiral Azo-polymeric domains. Furthermore, the circular dichroism of achiral Azo-polymers was found to increase when annealing the film above *T*_g_. In addition, the polarization grating surface relief gratings in Azo-polymers films with phototunable supramolecular helix pitch can be induced by chiral molecules. The helix pitch, thickness and LC phases significantly affect the induced supramolecular chirality and photo-optical properties of the achiral Azo-polymers.

Chiral low-molecular weight gelators can be regarded as a special chiral additive that can induce the self-assembly of achiral polymers into ordered supramolecular gelation nanostructures by sol-gel transition [79]. Zhang and Liu et al. reported an easily accessible gelation-guided supramolecular self-assembly system where the chirality of gelators (LBG-18 and DBG-18 in Figure 8) was easily transferred to the achiral Azo-polymers 17 [194]. Meanwhile, the supramolecular chirality and the helical handedness of achiral Azo-polymer 17 was entirely dominated by the molecular gelator chirality. More importantly, the lengths of the alkyl chain, the content of chiral gelator and photo irradiation can be varied to control the induced supramolecular chirality of achiral Azo-polymer 17.

Recently, Yu et al. fabricated helical topological structures of achiral Azo-copolymer 18 using chiral molecules (L-tartaric acid and D-tartaric acid in Figure 8). The enantiopure tartaric acid interacted with both blocks of Azo-polymers, resulting in helical morphologies upon supramolecular self-assembly. In this case, the chirality of chiral molecules was transferred and magnified in achiral polymer phase domains through supramolecular interactions, thereby inducing unique helical structures [195]. In addition to forming supramolecular gels, it should be noted that the chiral oligomers/polymers, termed as “scaffolds” or “templates”, can also be used as the chiral additives to induce the supramolecular helicity/chirality of achiral substances [196]. In 2019, Zhang and Fujiki et al. employed (poly(*n*-hexyl-(*S*)-2-methylbutylsiane), PSi-*S*, and poly(*n*-hexyl-(*R*)-2-methylbutylsiane), PSi-*R*) as chiral scaffolds to co-assemble with achiral main-chain Azo-polymer 19 [83]. In this case, aggregation-induced circular dichroism based on the co-assembly of chiral PSi with Azo-polymer 19 have been observed. The results showed that the induction effects of the chiral PSi scaffolds can be comparable to that of the chiral gelator, but the chiral PSi scaffolds can be more easily removed by photo-scissoring PSi by 313 nm light irradiation. Indeed, many supramolecular chiral assemblies are obtained by the co-assembly of achiral molecules and chiral additives; however, the chirality induction of achiral polymers based on Azo chromophores by chiral additives still requires further attention.

#### 3.3.2. Supramolecular Chirality Induced by Chiral Solvent

In addition to the induction strategies above, chiral solvation has been proven to be a universal and effective method for constructing supramolecular chiral superstructures from achiral polymer building blocks [197], such as polyacetylenes [198], polysilanes [68], poly(*n*-hexyl isocyanate) [199] and polyfluorene analogs [75,200]. Green et al., for the first time, reported that one-handed helical poly(*n*-hexyl isocyanate) can be obtained when the racemic polymers were treated with a variety of chiral solvents [199]. This outstanding work realized the possibility of transferring chirality from chiral solvents to the achiral polymer structures.

Our group investigated the details of induction of supramolecular chirality in achiral side-chain Azo-polymers by chiral limonene. In 2015, we presented the first construction of supramolecular chirality based on achiral Azo-polymer 20 (Figure 9) by supramolecular self-assembly using chiral limonene simultaneously as the chiral solvent and poor solvent. The induced chirality in the polymer was found to be a result of supramolecular chirality of the aggregation by the π-π stacking of achiral Azo building blocks in polymer side chains (Figure 10) [135]. Furthermore, the polymer structures and volume fractions of each solvent (good solvent and chiral (poor) limonene) have an obvious influence on the ordered chiral structures. More interestingly, the supramolecular chirality can be easily controlled by UV light and heating-cooling treatment, which was ascribed to the *trans*-*cis*-*trans* isomerization of Azo chromophores. In this system, chiroptical switching was achieved through the reversible modulation of the supramolecular helicity of Azo building blocks. Subsequently, Zhang and coworkers systematically investigated the effects of spacer lengths, topological structures and push-pull electronic substituents in Azo-polymer 20, on the chiral limonene induced supramolecular chirality [133,134,201]. The results of CD and UV-vis spectra suggested that the internal structures and external stimuli have distinctive influences on the induction and regulation of supramolecular chirality of achiral Azo-polymers. Recently, we used the same Azo-polymer 20 and successfully yielded CD-active Azo-polymer films triggered by chiral limonene vapor [137]. Chiroptical induction, switching, memory of the polymer films were systematically studied. The proposed mechanism was the preferred π-π helical stacking of achiral Azo units, similar to the previous works.

#### 3.3.3. Supramolecular Chirality Induced by CPL

It has been reported that circular polarization plays an important role in inducing chiral asymmetry in interstellar organic molecules, which might be observed in sunlight. CPL has been considered to be the plausible origin of the single handed homochirality of biomolecules and could explain the excess of L-amino acids found in nature [202]. Essentially, the circular polarization of light is the rotation of electric field waves in a plane perpendicular to its direction of propagation while maintaining a constant amplitude over time. The right-CPL (*r*-CPL) and left-CPL (*l*-CPL) can be easily obtained by adjusting the angle between the Glan-Taylor prism and 1/4 quartz waveplate (45^o^ or 135^o^). Among the different methods for constructing supramolecular chiral structures from achiral Azo-polymers, CPL has been used to provide an additional spatiotemporal control for the asymmetric properties of the polymers. At the same time, photosensitive Azo units can be used as an intermediate medium to transfer chiral information from CPL to the polymers.

Actually, CPL has become a well-accepted tool to control the chirality of polymers since the first demonstration of CPL-induced chirality in achiral Azo-polymers from Nikolova’s group. In 1997, Nikolova et al. reported that large circular birefringence and circular dichroism could be observed in achiral LC Azo-polymer 21 (Figure 11) films by illumination with one-handed CPL. In addition, the handedness of the CPL determines the sign of circular optical property of the polymer film [203]. A strong optical activity in another achiral LC Azo-polymer 22 upon illumination with CPL was subsequently reported in 1999 [204]. The proposed mechanism for the formation of a new ordered chiral structure when the polymer film was irradiated by CPL, was that angular momentum from the CPL was transferred to the Azo chromophore and the CPL induced the reorientation of the Azo chromophores. Nikolova et al. systematically investigated the structural effect on the photoinduced chirality of achiral amorphous and LC Azo-polymers (21 and 23) to clarify the mechanism [205]. The results showed that different values of the output polarization azimuth in the light polarization rotation between these two types of Azo-polymers were responsible for the induction of chiral structures. They believed the mechanism may be that a large decrease of the output polarization azimuth was induced for larger input ellipticity in achiral amorphous Azo-polymer 23. However, in the case of achiral LC Azo-polymer 21, no such reduction was observed. Due to the thickness of the polymer film, the optical axis gradually rotates as the light propagates through the sample. This may be the reason for the chiral orientation of the Azo chromophores induced by CPL. The first layer of the films changes the polarization of the initial CP light to an elliptical state (Elliptically polarized light, EPL, Figure 12). The EPL continues to change along successive film layers of in a way analogous to that in the amorphous polymer film, in which Azo chromophores also undergo angular reorientation at the same time [206]. It should be pointed out that the change of ellipticity may not be linear along the thickness of the polymer film, since each anisotropic domain of the LC Azo-polymers has its own orientation of the director. The reorientation of the Azo units induced by CPL leads to an organized helical arrangement via the aggregation of side-chain Azo units, thereby causing the chirality of achiral Azo-polymers.

Many excellent works on the CPL-induced supramolecular chirality of Azo-polymers have been published. For example, Iftime et al. firstly reported that CPL with opposite handedness produced a one-handed supramolecular helical structure for achiral Azo-polymer 24 in the smectic phase [207]. The handedness of the supramolecular helical structure was dependent on the handedness of the CPL. The mechanism of the supramolecular structure generated by CPL source may be the formation of a planar TGB phase. Furthermore, supramolecular chirality also can be constructed in amorphous polymers (such as Azo-polymer 25, reported by Ivanov et al.), but the polymers need to be pre-oriented in these cases [208,209,210,211]. However, Kim et al. reported that EPL successfully induced the chirality of achiral amorphous Azo-polymer 26 films, but this was not achieved with CPL [212,213,214]. The results suggested that liquid crystallinity is not a necessary condition for Azo-polymers to exhibit photoinduced chirality. We can draw a conclusion from the above reports that chirality can be induced by CPL in achiral LC Azo-polymers, or in achiral amorphous Azo-polymers with the preorientation of Azo units. The chirality can also be directly induced by EPL without preorientation in both achiral LC and amorphous Azo-polymers. Nevertheless, Cipparrone et al. reported that CPL can directly induce supramolecular chirality even in an achiral amorphous Azo-polymer 27 film without any preorientation [36,215,216,217]. Even though the induced mechanism of achiral amorphous Azo-polymers is still not clearly understood, which presents a new area of research in this field.

Natansohn and Rochon et al. presented many pioneering works on the photoinduced chirality of achiral Azo-polymers [218,219,220,221]. In 2004, they found that circular dichroism of Azo-copolymer 28 can be induced by CPL due to the cooperative motion between Azo units and nonphotoactive mesogens during the process of supramolecular self-assembly. Oriol et al. made in-depth investigations on CPL-induced supramolecular chirality in achiral Azo-homopolymers/copolymers 29 [131,222,223,224,225,226,227,228]. Zhang et al. systematically studied the effects of different spacers, the type of Azo chromophore and end groups on the CPL-induced chirality of achiral LC Azo-polymers 30 [229,230]. Many researchers have done a lot of works on CPL-induced supramolecular chirality and believed that CPL sense (*r* or *l*) can effectively control the chirality of achiral Azo-polymers. However, the dependence of photogeneration, photoresolution and photoswitching modes of achiral side-chain Azo-polymers on the CPL wavelength were rarely investigated. Recently, Zhang and Fujiki et al. demonstrated that the wavelength of the CPL source (313 nm and 365/405/436 nm) is the determining factor for photogeneration, photoresolution, and photoswitching of the achiral Azo-polymer 20 aggregates [231], regardless of the CPL sense (Figure 13). Completely symmetrical bisignate Cotton effects at the π-π* transitions of Azo chromophores were observed in the CD spectra of Azo-polymer aggregates upon *l*- and *r*-CPL (365 nm) irradiation. Interestingly, the CD sign is completely reversed when the Azo-polymer 20 aggregates were irradiated by 313 nm *l*- and *r*-CPL. The opposite supramolecular chirality was observed after the irradiation of CPL with different wavelengths. These results provided a new insight into the CPL-induced and controlled chirality of molecules, supramolecules, and polymers which have enormous potential in the mirror symmetry breaking field.

## 4. Supramolecular Chirality of Azo-Polymers Constructed by In situ Supramolecular Self-Assembly

Supramolecular chirality of Azo-polymers produced from the self-assembly of chiral Azo units or achiral Azo units induced by chiral sources has been deeply investigated in solution and solid film. As described above, the postpolymerization supramolecular self-assembly system generally contains two steps, in other words, the corresponding polymers need to be synthesized first, and then self-assembled in an asymmetric manner. Besides, the postpolymerization supramolecular self-assembly is a relatively tedious strategy for producing supramolecular chiral assemblies, which usually require a complex system consisting of the good solvent, poor solvent and chiral solvent. Furthermore, this traditional strategy is normally performed at rather low concentrations, which limits many potential applications. Generally, supramolecular chiral assemblies of Azo-polymers obtained by this strategy always contain irregular aggregates and they are difficult to control in solution. Therefore, whether and how particular morphologies of assemblies affects the chiral expression is poorly understood and worth exploring.

Over the past two decades, PISA has been developed as a versatile route to construct phase-separated assemblies. PISA has emerged as a convenient method for manufacturing the desired morphologies which do not require any post-polymerization processing. Compared to postpolymerization self-assembly, the essence of PISA is that the self-assembly is simultaneously accomplished and modulated during the process of polymerization. During a typical RAFT-mediated PISA, co-monomers are added to a homopolymer to extend the chain to yield asymmetric block copolymers. During the chain extension with the co-monomer, the solubility of the second block in polymerization media decreases, driving the self-assembly of block copolymers into nanoaggregates and changing their morphological phases. Many excellent works have been reported about Azo-polymer assemblies constructed by PISA strategy. For example, in 2018, Chen et al. reported the scale-up formation of Azo-polymer assemblies with anisotropic morphologies by PISA in ethanol (Figure 14a) [232]. Various anisotropic assemblies, including cuboids, short belts, lamellae and ellipsoidal vesicles, were obtained in a remarkably broad range of phase diagram. The morphologies of anisotropic structures are regulated by the balance of the surface energy and internal LC packing energy. The proposed mechanism is that the internal Azo LC ordering and the self-assembly of the amphiphilic Azo block copolymer occurred simultaneously, to provide the controlled construction of hierarchical Azo-polymer assemblies. This work emphasized the first large-scale synthesis of LC Azo-polymer nanocuboids, which may be useful for applications as actuators and drug delivery systems with photo control. Based on the advantages of PISA strategy, a light-triggered reversible slimming phenomenon of Azo-containing wormlike nanoparticles was also demonstrate (Figure 14b) [233]. The pure worm phase covered a remarkably broad region in the morphological phase diagram because of the rather rigid nature of the Azo core-forming block. Worms can be expanded or slimed under alternating UV/visible light irradiation due to the photoisomerization property of Azo chromophores. These photoresponsive wormlike assemblies can be designed as nanofiltration switches, displaying variable rejection performance for small molecules upon alternative UV/visible light irradiation. Armes et al. used PISA strategy as well as host-guest chemistry to prepare Azo-containing worms (Figure 14c) [234]. The binding of an Azo-mPEG analyte to *β*-cyclodextrin (*β*-CD) functionalized vesicles prepared by PISA leads to a significantly faster morphology change. Moreover, the rate of such morphological transitions can be further fine-tuned via the photoisomerization of Azo chromophores. By the combination of PISA strategy and *trans*-*cis* photoisomerization of Azo chromophores, Yuan et al. achieved the reversible worm-to-vesicle transformation under alternating UV/visible light irradiation (Figure 14d) [235]. More interestingly, a series of intermediate morphologies of Azo-polymers, including coalesced worms, as well as “octopus”-like and ”jellyfish”-like structures, were revealed in the photoisomerization process.

Polymeric supramolecular assemblies can provide simple and versatile approaches for fabricating specific nanostructures and functional nanocomposites. The recent application of PISA strategy to construct polymeric supramolecular self-assemblies in situ has improved the chemical and physical properties of these new materials. PISA strategy provides opportunities and efficiency for preparing supramolecular assemblies both in material science and supramolecular chemistry. Through the PISA-mediated in situ supramolecular self-assembly, Gao et al. reported a supramolecular self-assembly of site-specific in situ polymerization induced self-assembly (SI-PISA), by combining site-specific in situ growth with PISA [236]. They, for the first time, synthesized therapeutic protein interferon-α micelle (IFN-micelles) by PISA, which has a significantly enhanced pharmacology for tumor therapy. More importantly, the remarkably enhanced therapeutic efficacy of IFN-micelles did not induce systemic toxicity. Furthermore, the dilemma that the half-life extension is usually limited by the decrease in bioactivity was addressed with the help of supramolecular self-assembly using SI-PISA. D’Agosto et al. presented that waterborne polymer assemblies bearing supramolecular association unit can be directly and efficiently synthesized by PISA [237]. The accessibility of supramolecular interaction between host and guest molecules proved that PISA even has great potential to turn nanospheres into building blocks for supramolecular structures. Recently, Rieger et al. proposed a templated-PISA strategy that allows for the directional and robust synthesis of supramolecular fibers [31]. The principle of this strategy was to introduce a hydrogen-bonded bis-urea sticker and utilize supramolecular interaction to drive the assemblies towards purely fiber morphology.

Chiral or helical polymeric supramolecular nanostructures have been extensively studied by researchers through traditional postpolymerization self-assembly method. However, few examples have been demonstrated for the construction of chiral polymer assemblies during the polymerization process. Until 2015, De et al. reported PISA of benzyl methacrylate (BzMA) using macro-CTA containing chiral side-chain amino acid as a solvophilic steric stabilizer, which allowed amphiphilic copolymers to self-assemble in situ into chiral nanostructures including spheres, worms, twisted fibers and vesicles, depending on the targeted block composition and the copolymer concentration (Figure 15a) [109]. Chirality transfer from the molecular level to the macromolecular level, and then to the supramolecular assemblies was observed by CD spectra and SEM images. The mechanism for such chirality transfer and twisted morphologies may be ascribed to hydrogen bonding between the side-chain chiral amino acids. To this end, main-chain helical assemblies have been constructed by PISA but the mechanism is distinctly different from that of supramolecular chirality in polymer side chains. In 2020, O’Reilly et al. presented a facile and tunable one-pot methodology, termed nickel-catalyzed coordination PISA (NiCCO-PISA) [238], to construct helical poly(aryl isocyanide)s in situ (Figure 15b). Various chiral superstructure morphologies, such as spheres, worms and vesicles, were constructed simultaneously during the process of polymerization. The helicity of the assemblies was detected via CD spectroscopy for all morphologies, proving the helical nature of their main chain formed in PISA process. Furthermore, Wan et al. recently reported that PISA strategy can also be employed to prepare main-chain helical assemblies of conjugated poly(phenylacetylene)s [239], which can self-assemble spontaneously into vesicles, lamellae and helical ribbons with the increase of core-forming blocks length (Figure 15c). The chiral properties and the formation of primary helices as main-chain twisted stacks were investigated by TEM and further confirmed by CD data. The results indicated that molecular chirality could be accurately transferred to the one-handed helical backbone of macromolecules, and then to the assemblies. These works not only expanded applications of PISA into conjugated polymers but also enhanced the preparation efficiency of chiral assemblies in situ.

According to the above reports, PISA has been shown to be a versatile protocol in constructing chiral assemblies in situ. However, the chirality of the above assemblies is derived from the helical structure of the polymer main chain, with little or no reference to the concept of supramolecular chirality. Recently, supramolecular chirality has been an emerging field of science that offers exciting possibilities for the assemblies with unprecedented features. Nevertheless, hierarchical Azo-polymer assemblies with supramolecular chirality are challenging to construct by in situ supramolecular self-assembly. Fortunately, our group, for the first time, reported a polymerization induced chiral self-assembly (PICSA) as a universal platform for the in situ controlled construction of hierarchical supramolecular chiral Azo-polymer assemblies (Figure 16a) [240]. The soluble poly(methacrylic acid) macro-CTA was chain-extended with chiral Azo monomers in ethanol to prepare a series of supramolecular chiral assemblies. Different morphologies were obtained by systematically tuning the length of Azo core-forming blocks. The morphology evolution of the assemblies was recorded by TEM and AFM images. A full phase diagram of morphologies spanning spheres, worms, helical fibers (Figure 16b), lamellae and vesicles were observed in a remarkably broad range of block compositions. Furthermore, the Azo-polymer assemblies exhibited a smectic phase with constant layer spacing from SAXS patterns. This internal LC ordering had a crucial effect on the formation and evolution of hierarchical morphologies. Apart from the simultaneous supramolecular self-assembly and internal Azo LC ordering, supramolecular chirality was also induced in Azo-BCP assemblies concurrently during the PICSA process. All morphologies were characterized by CD and UV-vis spectroscopies. Bisignate Cotton bands at the π-π* transitions of Azo chromophores were observed as intense couplets coupling in the CD spectra of Azo-polymer assemblies, confirming the chirality transfer from point chirality to Azo LC domains, and then to the supramolecular assemblies. The supramolecular chirality in the assemblies found to be a result of π-π stacking of Azo units. More importantly, different morphologies showed different chiral expression capabilities. For example, the CD intensity increased first and then decreased with the morphological transition from spheres to worms, lamellae and vesicles (Figure 16c). These results indicated that the morphological transition has crucial effects on the induction and expression of supramolecular chirality during the PICSA process, which cannot be achieved in the postpolymerization supramolecular self-assembly of Azo-polymers. From this study, we can conclude that PICSA as a new, general, and effective platform will have broad applications for the controlled construction of supramolecular chiral assemblies in situ and on a large scale.

## 5. Theoretical Calculations and Simulations

In addition to the described strategies and techniques that constructing and analyzing supramolecular chirality in Azo-polymers, multiscale molecular models and computer simulations offer a support for the combination of practice and theory in this field. Computer calculations and simulations can help in understanding the self-assembly process and allow studying the key interactions that control the supramolecular chiral structures and its intrinsic dynamics properties. There are many examples of using Molecular Dynamics (MD) simulations to study the assembly manner of Azo units [241,242]. For example, the *trans*-*cis* isomerization in the Azo units can generates distortions and changes in the properties of the entire supramolecular structures [243,244,245]. In this case, the effect of light on the supramolecular assemblies can be modelled at atomistic level by using MD simulations [246]. Klajn et al. modified the –C–N=N–C– dihedral angles in Azo units to make them tend to undergo *trans*-*cis* transition autonomously during a MD simulation [247]. Cho et al. developed a multiscale analysis framework based on coarse-grained MD simulations to construct the Azo cross-linked liquid crystalline polymers networks model that satisfies the structural and thermodynamic properties of the full-atomistic reference [248]. Employing computer simulation techniques, Guskova et al. have analyzed the noncovalent interactions between the Azo arms in benzene-1,3,5-tricarboxamide columns [249,250]. Furthermore, Kudernac et al. used all-atom MD method to simulate the effect of *trans*-*cis* isomerization on the self-assembled nanotube formed by angular monomers containing Azo tails [251]. The computer simulations demonstrated that the structure of supramolecular assembly was damaged over a certain threshold during the *trans* to *cis* transition of Azo units, which is consistent with the experimental evidence. For the theoretical calculation and simulation of the supramolecular chirality of Azo-polymers, Zhang and Fujiki et al. used ZINDO calculations (Zerner’s Intermediate Neglect of Differential Overlap) to verify the origin of the bisignate cotton CD bands of achiral Azo-polymer 20 (x = 6; R = OCH_3_), as described above [231]. The most favorable assorted structures of Azo side chains tend to adopt chirally tilted π-π stacking motifs (Figure 17), which may be assumed to be responsible for the bisignate cotton CD band that was observed experimentally. And the opposite cotton bands at the long axis (∼400 nm) and short axis (∼300 nm) may be the origin of the wavelength dependence of CPL-driven chiroptical generation, swapping and inversion of the Azo-polymer 20 aggregates.

## 6. Conclusions and Outlook

Supramolecular chirality is one of the most basic concepts of modern chemical science and has many indispensable functions in living organisms and nature. The induction and construction of supramolecular chirality in Azo-polymers combines the advantages of fascinating supramolecular chiral superstructures and photo-responsive properties of Azo-polymers. The supramolecular chirality in Azo-polymers is mainly constructed through the asymmetric arrangement of noncovalent Azo building blocks, using either LC or amorphous Azo-polymers. Generally, there are two common methods for producing nanoassemblies in the supramolecular self-assembly of Azo-polymers. The first one is postpolymerization self-assembly, which requires multi-step procedures. The other is in situ supramolecular self-assembly, which has been greatly developed due to the proposal of PISA concept. This review demonstrates recent progress in the field of supramolecular chirality in Azo-polymers with a particular emphasis on the advantages of an in situ supramolecular self-assembly strategy to chirality induction and construction. In either the solid film, dissolved state or solution aggregates of Azo-polymers, supramolecular chiral superstructures can be effectively constructed in Azo-polymer systems through above two methods. However, despite the novelty of these superstructures and apparent attractiveness of PISA methodology, the detailed nature of the chirality-transfer mechanisms and direct visualization of in situ supramolecular self-assembly process become imperative. In addition, increased attention should be focused on the universality of the in situ supramolecular self-assembly, especially PICSA strategy, and the controllable preparation of supramolecular chiral Azo-assemblies with inverse bicontinuous mesophases. Moreover, further efforts should be concentrated on developing applications of supramolecular chirality of Azo-polymers, such as chiral recognition, chiral switching, electro-optics and biological fields. Although there are still many unknowns related to the supramolecular chirality of Azo-polymers, we hope that this review can attract more interest from researchers and further advance the development of the construction of supramolecular chirality in Azo-polymers.

## Figures and Tables

**Figure 1 ijms-21-06186-f001:**
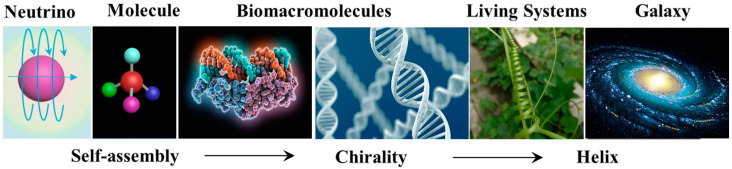
Chiral architectures at various scales, from neutrino to enantiomeric molecules, proteins and DNA biomacromolecules, macroscopic living systems and galaxy.

**Figure 2 ijms-21-06186-f002:**
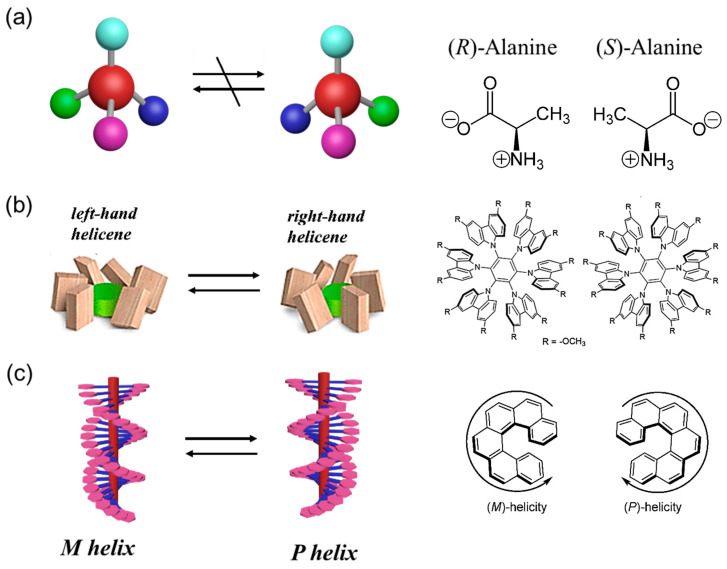
Schematic representation of chiroptical switching between two enantiomers possessing (**a**) configurational chirality, (**b**) conformational chirality and (**c**) helical chirality.

**Figure 3 ijms-21-06186-f003:**
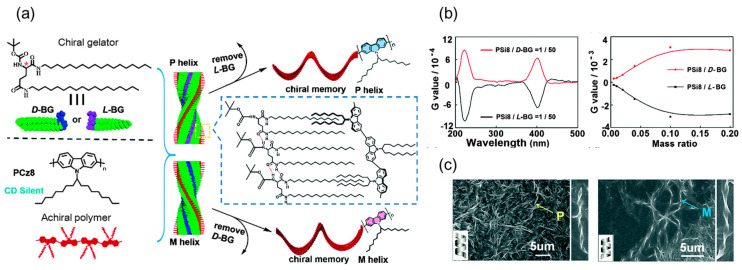
(**a**) Illustration of chiral transfer from gelator to achiral polymer. (**b**) CD spectra of the supramolecular chiral co-gels. (**c**) SEM images of the helical fibers. (Reproduced from [79] with permission from The Royal Society of Chemistry).

**Figure 4 ijms-21-06186-f004:**
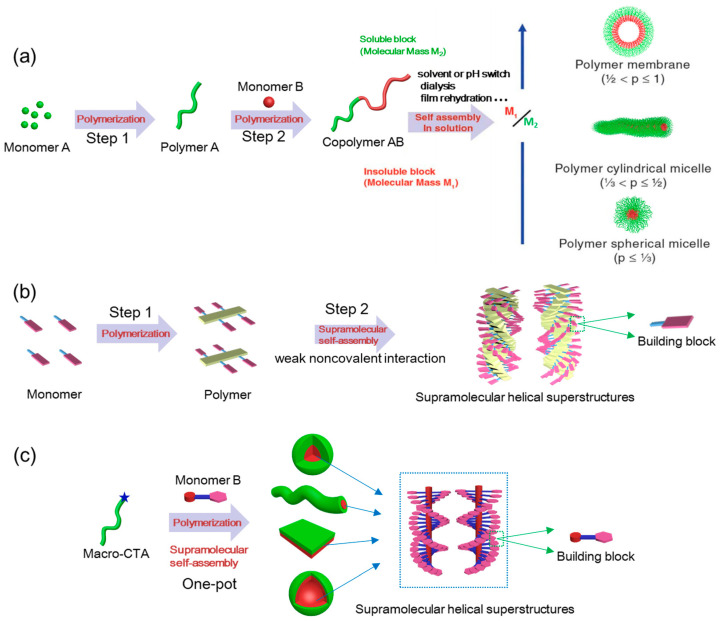
Schematic representation of (**a**) traditional postpolymerization self-assembly, (**b**) previous supramolecular self-assembly in polymer systems and (**c**) in situ supramolecular self-assembly mediated by PISA.

**Figure 5 ijms-21-06186-f005:**
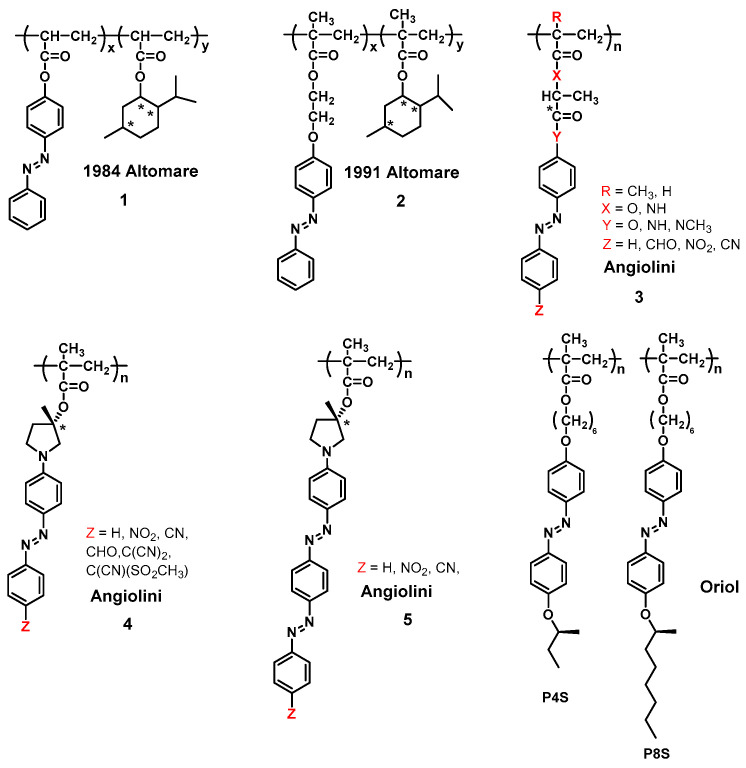
The chemical structures of chiral Azo-polymers 1, 2, 3, 4, 5, P4S and P8S. (R, X, Y, and Z represent different groups).

**Figure 6 ijms-21-06186-f006:**
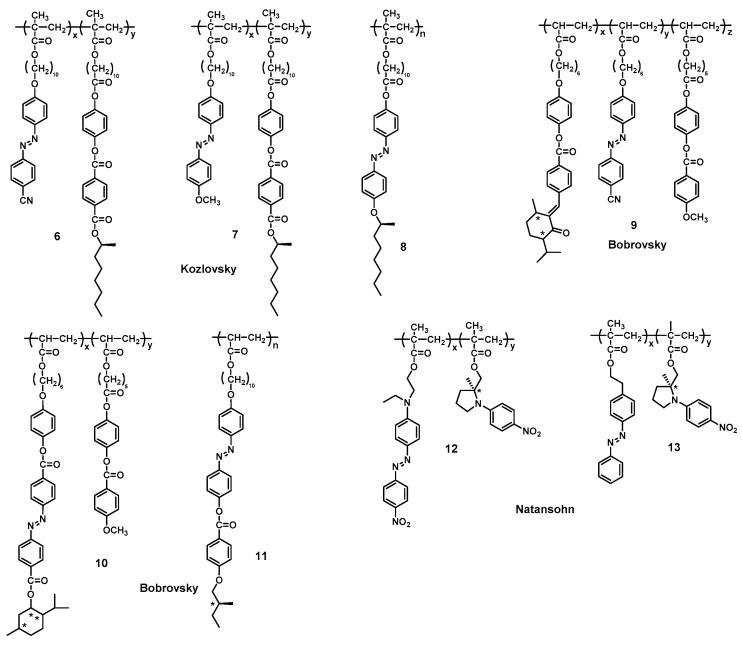
The chemical structures of chiral Azo-polymers 6, 7, 8, 9, 10, 11, 12 and 13.

**Figure 7 ijms-21-06186-f007:**
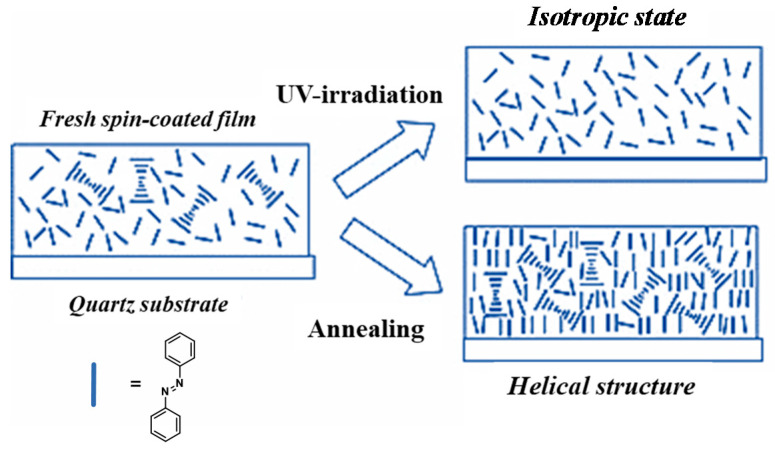
Idealized scheme of photoinduced and thermoinduced structural changes in the spin-coated film of the chiral Azo-polymer 11.

**Figure 8 ijms-21-06186-f008:**
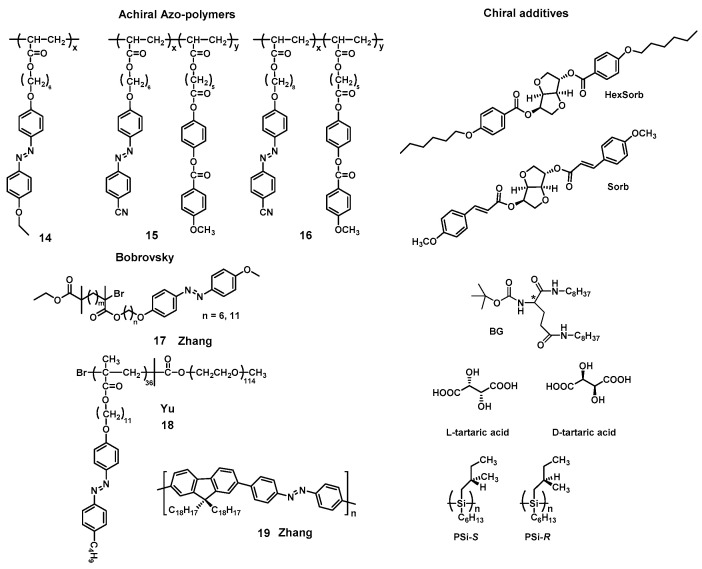
The chemical structures of achiral Azo-polymers 14, 15, 16, 17, 18, 19 and chiral additives.

**Figure 9 ijms-21-06186-f009:**
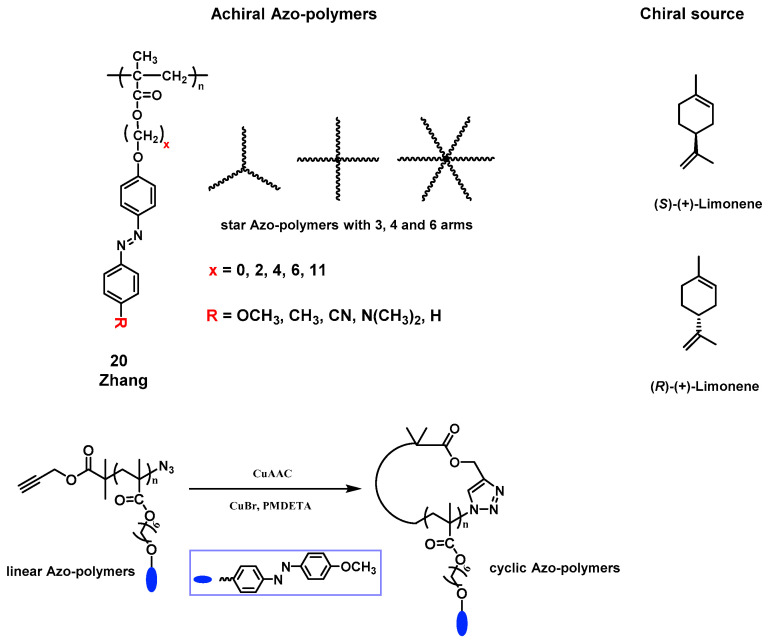
The chemical structures of achiral Azo-polymers based on Azo-polymer 20 and chiral limonene. (x represents different spacer lengths; R represents different substituents).

**Figure 10 ijms-21-06186-f010:**
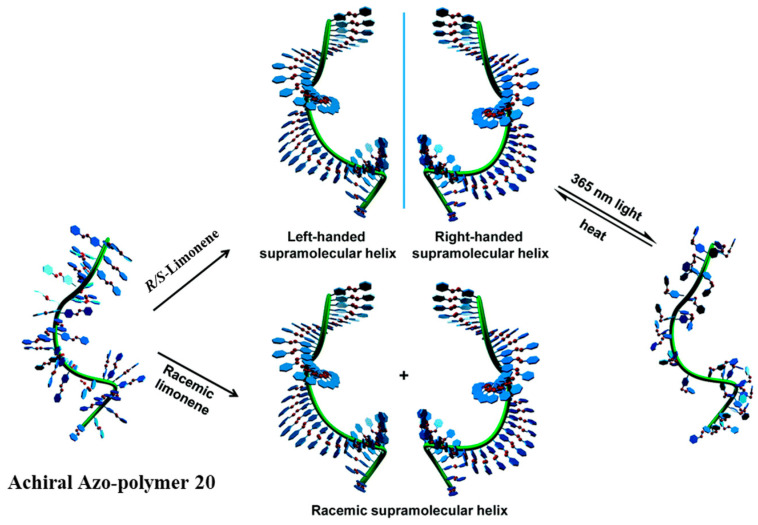
Illustration of the supramolecular helical structures of achiral Azo-polymer 20 induced by chiral limonene and the construction of the chiroptical switching. (Reproduced from [135] with permission from The Royal Society of Chemistry).

**Figure 11 ijms-21-06186-f011:**
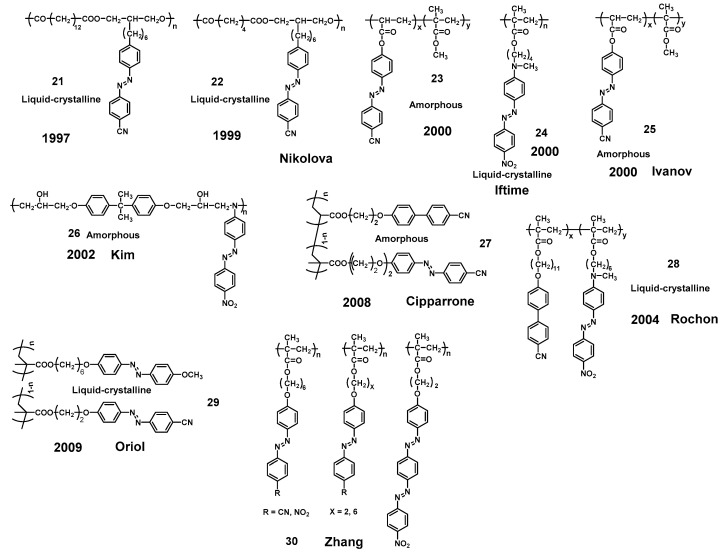
The chemical structures of achiral Azo-polymers 21, 22, 23, 24, 25, 26, 27, 28, 29 and 30.

**Figure 12 ijms-21-06186-f012:**
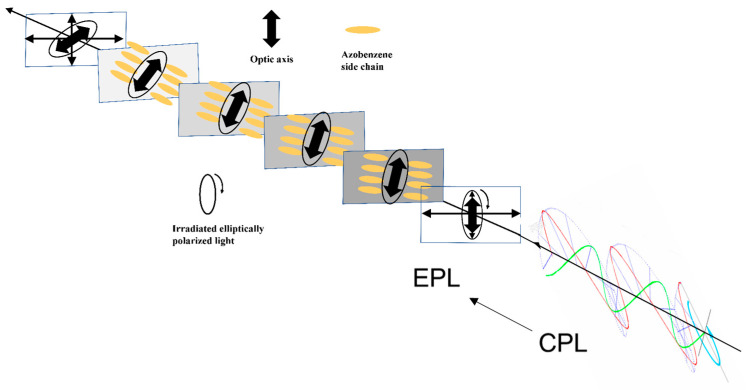
Sequence illustrating the formation of helical structure by CPL irradiation. (Reproduced from [206] with permission from The Royal Society of Chemistry).

**Figure 13 ijms-21-06186-f013:**
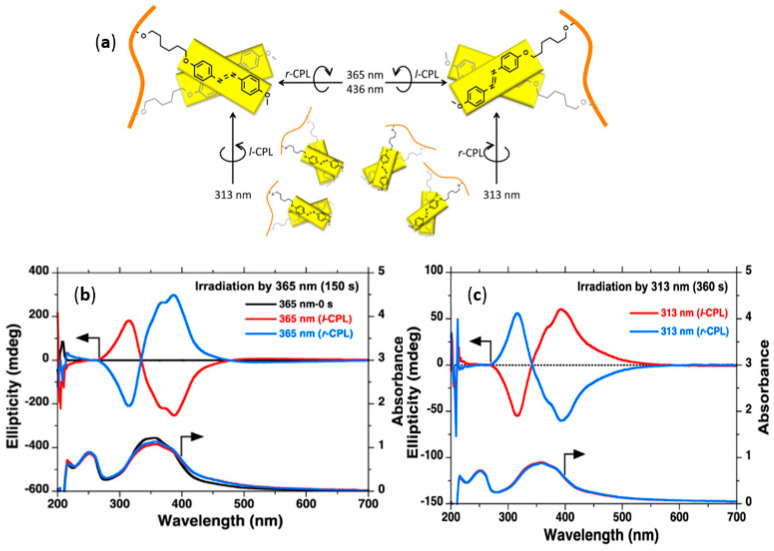
Proposed scheme of twisted stack led by wavelength-dependent (**a**) l- and r-CPL sources. CD and UV-vis spectra of Azo-polymer 20 aggregates exposed to (**b**) 365 nm and (**c**) 313 nm l-CPL and r-CPL. (Reprinted with permission from [231]; Copyright (2017) American Chemical Society).

**Figure 14 ijms-21-06186-f014:**
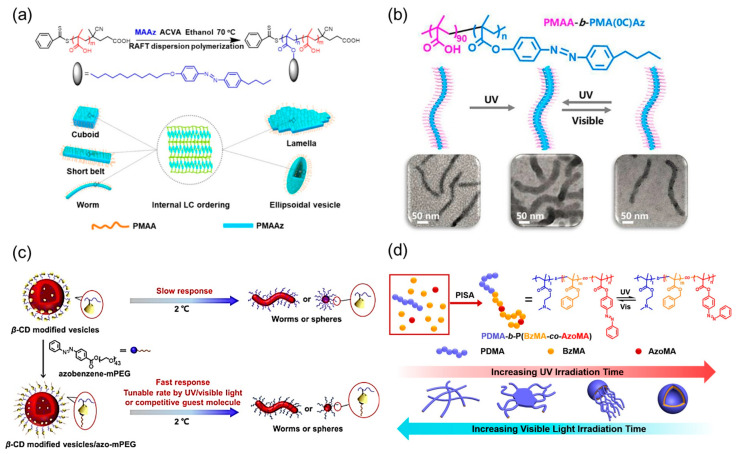
Schematic representation of Azo-polymer assemblies and morphological transition constructed by PISA strategy. (For (**a**–**d**), Reprinted with permission from [232,233,234,235]; Copyright (2018, 2019, 2017, 2018) American Chemical Society).

**Figure 15 ijms-21-06186-f015:**
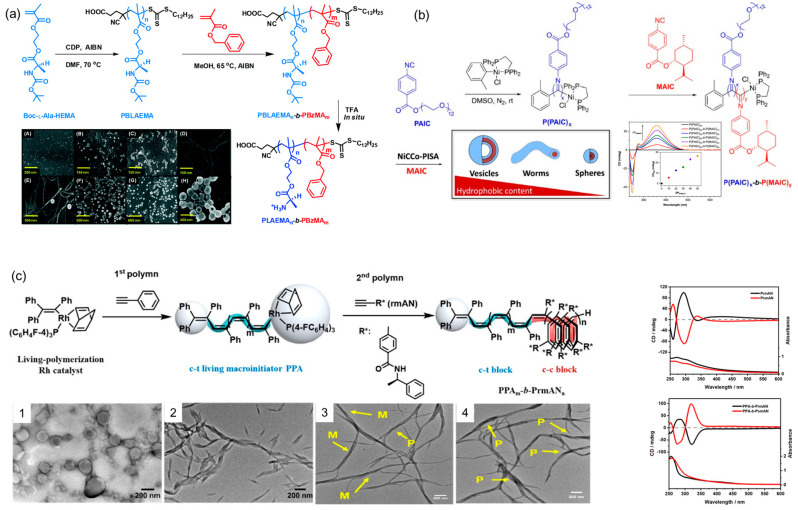
Scheme illustrating the construction of chiral or helical assemblies based on PISA strategy. (Red represents solvophobic segments, and blue represents solvophilic segments). ((**a**) Reproduced from [109] with permission from The Royal Society of Chemistry; (**b**) and (**c**), reprinted with permission from [238,239]; Copyright (2020, 2020) American Chemical Society).

**Figure 16 ijms-21-06186-f016:**
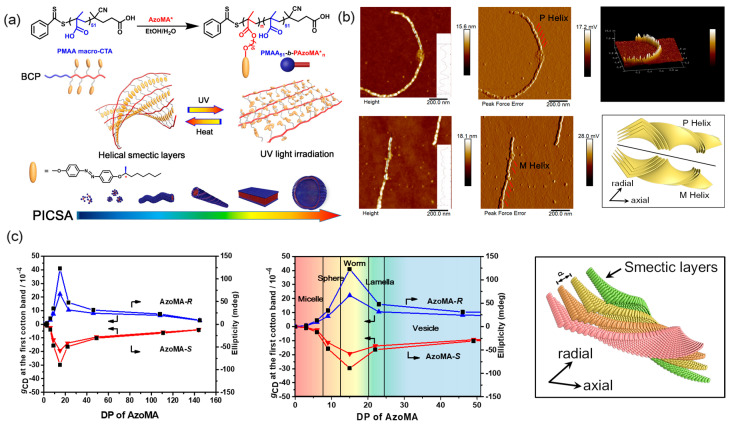
(**a**) Synthetic route of PICSA strategy for constructing hierarchical supramolecular chiral Azo-polymer assemblies. Inset: representation of the dynamic reversibility. (**b**) AFM images and the representation of the helical fibers. (**c**) The maximum CD and *g*_CD_ values of the supramolecular chiral Azo-polymer assemblies with different morphologies and the representation of the supramolecular helical structure.

**Figure 17 ijms-21-06186-f017:**
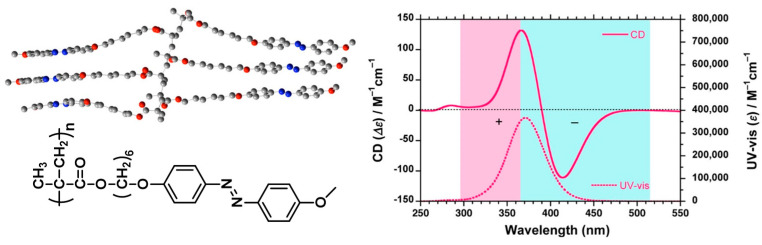
Six slightly twisted *trans*-Azo side chains and simulated CD and UV-vis spectra of the model. (Reprinted with permission from [231]; Copyright (2017) American Chemical Society).

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
