# Peer review of "Supramolecular Chirality in Azobenzene-Containing Polymer System: Traditional Postpolymerization Self-Assembly Versus In Situ Supramolecular Self-Assembly Strategy"

_ijms, 2020, doi:10.3390/ijms21176186_

Round 1

Reviewer 1 Report

This manuscript reviews chiral molecular and polymer systems where azobenzene’s transformation between its trans and cis
forms plays important roles. While the manuscript focuses on the
authors’ own research work, it also covers related, important papers.
I feel that the manuscript is suitable for publication in IJMS on an
as-is basis.

Author Response

Many thanks for your valuable comments.

Reviewer 2 Report

The authors review the supramolecular chiral organization of azo-polymer, comparing self-assembly postpolymerization and the in situ supramolecular self-assembly strategy. The work is well structured, clearly written and the bibliography is appropriate. However, the authors have to revise some important aspects before International Journal of Molecular Sciences publishing this review.

  1. There is a somewhat imprecise use of the term self-assembly, at certain times it seems that they refer to self-assembly for the formation of nano-objects (micelles, vesicles, ...) and at others to the formation of a supramolecular organization that does not necessarily have to be expressed as a nano-object. This imprecise use, repeated all over the text, should be corrected
  2. Line 88. Ref. 30 refers to PISA.
  3. Line 118. Work of Lehn et al should be cited.
  4. Line 127, “and” must be deleted.
  5. Lines from 192 to 209. The terms “host” and “guess” are rarely used in this context. Host-guest complexes, although based on non-covalent interactions, suggest molecular recognition and a unique structure relationship. Please, review this paragraph.
  6. Line 218. Delete the expression “supramolecular systems are generally nor crystallized”, it is not correct.
  7. Lines from 214 to 242. The description of the characterization techniques of chiral supramolecular organizations should be improved. The entire paragraph should be rewritten. The techniques should be described correctly. Details of the relevant information is obtained from each should be included. The order of presentation of the techniques is not correct. The morphological characterization should be presented at the end, the chiral supramolecular order does not always translate into a chiral morphology. The description of CD and VCD is imprecise, both CD and VCD spectrometers register the absorption difference of electromagnetic radiations, the difference between them is the frequency range in which they work.
  8. Figure 4. As I said before, it is not clearly explained what the authors refer to with self-assembly, only to the formation of nano-objects?
  9. Line 311. E/Z isomers are configurational isomers.
  10. Line from 312 to 326, should include the citations of the works that are described.
  11. Line 348, delete CHCl3.
  12. Line 430. J. del Barrio, R. M. Tejedor, L. S. Chinelatto, C. Sánchez, M. Piñol and L. Oriol, Chem. Mater., 2010, 22, 1714–1723 should be cited in this paragraph.
  13. Line 509 and following. A review should not be written in the first person. Really, the works of the authors are significant, but it is not necessary such exhaustive information compared to other interesting works cited.
  14. Line 597. J. Royes, C. Provenzano, P. Pagliusi, R. M. Tejedor, M. Piñol and L. Oriol, Macromol. Rapid Commun., 2014, 35, 1890–1895 should be cited.
  15. Line 598. R. M. Tejedor, L. Oriol, J. L. Serrano, F. Partal Ureña and J. J. López González, Adv. Funct. Mater., 2007, 17, 3486–3492 and J. Royes, V. Polo, S. Uriel, L. Oriol, M. Piñol, R. M. Tejedor, Phys. Chem. Chem. Phys., 2017, 19, 13622–13628 should be cited and the mechanism of the photoinduction of chirality discuss.
  16. Line 604. J. Royes, L. Oriol, R. M. Tejedor and M. Piñol, Polymers (Basel), 2019, 11, 885 should be cited.
  17. Line 632. And the circularly polarized light?
  18. Ref 57 should be corrected.

Reviewer 3 Report

This review summarizes 230 publications that describe basic concepts as well as advance experimental research in the field of supramolecular chirality of Azo-polymer systems. It also provides a wide perspective on recent trends in this growing field. I find the topic very interesting, the writing style is clear and fluent, the paper structure is logical and overall the review is highly appropriate for publication in International Journal of Molecular Sciences. Several minor issues do need corrections:

  1. On line 181 the authors state that: "In general, supramolecular chirality largely depends on the spatial arrangements or the assembly manner of the chiral/achiral building blocks. It not only requires weak interaction between the assembled components, but also a mechanism for asymmetric stacking, i.e. (1) one or more of the components are asymmetric that the assembly has determined supramolecular chiral structures; (2) or the achiral components associate in noncovalent interactions that the assembly has no elements of symmetry."

    While I agree with the first sentence of this paragraph, the use of the term asymmetry in the rest of the paragraph is inaccurate – while an asymmetric object is chiral, a chiral object does not have to be asymmetric. Molecules belonging to e.g., a Cn point group are both symmetric and chiral. A common example is the helix which is both chiral and C2-symmetric. Asymmetry means no symmetry at all, that is the object belongs to the C1 point group.
  2. On line 205 – Please consider adding another interesting application of inducing chirality on a polymer by creating a complex with a DNA. See for example Fossépre et al., ACS Appl. Bio Mater. 2019, 2, 2125−2136.
  3. On line 375 the authors state that: "In contrast to the conformational chirality reported by Angiolini, the chirality of P8S in aggregated state is associated with a supramolecular chiral assembly of Azo chromophores because the chiral terminal alkyl chain has a low influence on the polymer backbone. In addition, supramolecular chirality of the P8S polymer aggregates in DCM/hexane solution can be disrupted by trans-cis photoisomerization of Azo units, which further confirmed the CD signals described above were from the supramolecular chirality of Azo stacks."

    It is not clear from this paragraph whether these are the findings of Oriol et al. who are cited before that paragraph, in which case clearer citation is missing, or whether these are the conclusions drawn by the authors based on their experience, in which case, evidence are needed to justify these statements.
  4. One final comment regards the general perspective presented in this paper which is mostly experimental. As such it describes novel techniques to synthesize and analyze azo-polymers and supramolecular chirality. Many derivatives of azo polymers are described, but the experimental findings often miss the ability to provide a comprehensive molecular explanation of why the particular polymer creates the chiral effect in the first place – what is the part played by each of the structural fragments that construct the supramolecular system in dictating the chirality? In this respect computational methods and particularly molecular dynamics can be of value, and provide information on the factors that control supramolecular polymer structure and dynamics and gain insight regarding their mechanism of action. I suggest adding a paragraph on the contribution of computational studies to this growing field as a well as a call for more computational research that would support experimental results, preferably with specific suggestions of open questions. See for example the papers by Savchenko et al., Molecules 2019, 24, 4387 and Moon et al., Macromolecules 2019, 52, 5, 2033–2049, as well as the review by Davide Bochicchio & Giovanni M. Pavan, Advances in Physics: X, 2018, 3:1, 1436408 for some recent examples.

Round 2

Reviewer 2 Report

No comments